# Determinants of Undernutrition among Children Admitted to a Pediatric Hospital in Port Sudan, Sudan

**DOI:** 10.3390/nu16060787

**Published:** 2024-03-10

**Authors:** Giulia Chiopris, Caterina Chiopris, Manuela Valenti, Susanna Esposito

**Affiliations:** 1Pediatric Clinic, Department of Medicine and Surgery, University Hospital of Parma, 43126 Parma, Italy; giulia.chiopris@gmail.com; 2Department of Political Economy and Government, Harvard University, Cambridge, MA 02138, USA; chiopris@g.harvard.edu; 3Emergency NGO Onlus, 20122 Milan, Italy; manuela.valenti@emergency.it

**Keywords:** breastfeeding, kwashiorkor, malnutrition, undernutrition, wasting

## Abstract

Severe acute undernutrition (SAU) is still a crucial global health issue in the 0–59 months population, increasing the risk of mortality as well as of long-term consequences. In Sudan, 3.3 million children suffered from acute malnutrition between 2018 and 2019. This study was planned to evaluate, in the area of Port Sudan, the prevalence of acute undernutrition after the COVID-19 pandemic and to identify the most important factors favoring the development of acute undernutrition. The available clinical records of all the under-five children (*n* = 1012) admitted to the Port Sudan Emergency Pediatric Hospital from 1 February 2021 to 31 January 2022 were analyzed. The presence of wasting and kwashiorkor was assessed and children were categorized according to age, gender, place of residence, main reason for hospitalization, and underlying comorbidities. Acute undernutrition was evidenced in 493 (48.7%) children. Of them, only 16 (3.2%) were diagnosed with kwashiorkor. Children with SAU had a higher prevalence of acute gastroenteritis (*p* < 0.05) and parasitosis (*p* < 0.05). Infants aged 0–6 months were those with the lowest risk of undernutrition, whereas those aged 7–12 months were those with the greater risk. In these patients, multivariate analysis revealed that SAU and MAU were 2.5 times (OR 2.51; 95% CI, 1.79–3.55) and 5.5 times (OR 5.56; 95% CI, 2.59–18.7) higher. This study shows that the area of Port Sudan is still suffering from an alarming prevalence of severe wasting and the risk of developing acute undernutrition seems strictly related to the introduction of complementary feeding and tends to reduce with increasing age. Measures already in place to prevent acute malnutrition should be reinforced with improvement of mother education on child feeding.

## 1. Introduction

Good nutrition is essential for children’s survival and ability to achieve their full potential in terms of physical and mental growth [1]. Undernutrition in the first 1000 days of life may lead to severe developmental problems that may significantly reduce quality of life, increase the need for frequent medical visits and hospital admissions, and cause early death. Recent epidemiological evaluations have shown that around 45% of global deaths of children aged 0–5 years were linked to undernutrition [2]. Addressing undernutrition during the first 1000 days requires a multifaceted approach that involves improving maternal nutrition, promoting exclusive breastfeeding, ensuring access to nutritious foods, and providing healthcare and nutritional education to mothers and caregivers [2]. Investing in interventions during this critical window of opportunity can have significant long-term benefits for individuals, communities, and societies as a whole.

Undernutrition may cause stunting and wasting [3]. Stunting is diagnosed when the height-for-age of a child is significantly lower than expected and is the effect of chronic undernutrition. Stunting is primarily attributed to inadequate nutrition, particularly during the prenatal period and the first two years of life. Poor maternal nutrition, inadequate breastfeeding practices, and lack of access to nutritious foods contribute to stunting. Children who are stunted may experience developmental delays, reduced cognitive function, and an increased risk of chronic diseases later in life. Stunting also affects overall productivity and economic potential, perpetuating cycles of poverty and inequality. Wasting is diagnosed when a child has a low weight-for-height and is the result of a short period of undernutrition. Evaluation of the prevalence of wasting in a pediatric population is the best method to measure recent nutritional imbalance resulting in undernutrition and the immediate impact of actions specifically planned to face undernutrition [3]. Clinically, wasting includes marasmus, in which undernutrition derives from both energy and protein low intake, and kwashiorkor, which mainly depends on protein deficiency. Wasting is typically a result of acute food shortages, inadequate dietary diversity, or illness such as diarrhea or infections. It can occur suddenly and requires immediate intervention to prevent further deterioration. Wasting poses immediate risks to a child’s health and survival. Severe wasting increases the risk of mortality, particularly in young children. Even if a child recovers from wasting, they may experience long-term consequences such as impaired growth and development. Addressing stunting and wasting requires a coordinated effort across multiple sectors, including healthcare, nutrition, agriculture, and education, to ensure that children have the best possible start in life. Wasting caused by factors other than inadequate intake alone is referred to as cachexia or anorexia and could be considered secondary malnutrition due to comorbidities [3].

Although in recent years the initiatives of several institutions have led to a significant reduction in child undernutrition and related problems in almost all the developing countries [4], the prevalence of poor child nutrition remains very high, particularly in some sub-Saharan and Asian countries. Sudan is one of the poorest sub-Saharan countries and one of those where pediatric undernutrition has the highest frequency. Data from the World Health Organization (WHO) Tracking Tools of the Global Nutrition Targets across 2018–2019 showed that in this country all indicators of child malnutrition still remained very high [4]. Globally, in 2022, 149 million children under 5 were estimated to be stunted, 45 million were estimated to be wasted (too thin for height), and 37 million were overweight or obese. Nearly half of deaths among children under 5 years of age are linked to undernutrition. These mostly occur in low- and middle-income countries. The developmental, economic, social, and medical impacts of the global burden of malnutrition are serious and lasting, for individuals and their families, for communities, and for countries [4]. Overall, the national rate of stunting prevalence in Sudan was at 36.35%, with moderate and severe cases at 21.25% and 15.06%, respectively [5]. Moreover, 13.6% of children were diagnosed with wasting and 10.8% and 2.7% had moderate and severe manifestations, respectively, with the highest values in the Red Sea State [5]. In Sudan, around 19% of children under 5 years were underweight, although there may be disparities in undernutrition’s prevalence between different regions within Sudan [5]. Conflict-affected areas, rural communities, and areas with limited access to healthcare and basic services often experience higher rates of undernutrition. Undernutrition in Sudan is influenced by various factors, including food insecurity, inadequate access to clean water and sanitation, limited healthcare services, poverty, and ongoing conflicts. Humanitarian organizations and government agencies have been working to address undernutrition in Sudan through various interventions, including providing therapeutic feeding programs, nutritional supplements, access to clean water, and healthcare services. The COVID-19 pandemic may have exacerbated existing challenges related to undernutrition in this country, including disruptions to food supply chains, healthcare services, and economic instability, further affecting vulnerable populations. Periodic monitoring of the prevalence of acute undernutrition is essential to evaluate the impact of interventions and to measure the interference of new negative factors such as the recent COVID-19 pandemic. This study was planned to evaluate, in the area of Port Sudan, the prevalence of acute undernutrition after the COVID-19 pandemic and to identify the most important factors favoring the development of acute undernutrition.

## 2. Methods

### 2.1. Study Design

The available clinical records of all the under-five children admitted to the Port Sudan Emergency Pediatric Hospital from 1 February 2021 to 31 January 2022 were analyzed. Emergency is an Italian nongovernmental organization that offers qualified medical assistance with guaranteed collection of reliable data [6]. The demographic and social environmental characteristics of the enrolled children were recorded, including previous medical history, age, body weight, height, reason for hospitalization with detailed presenting symptoms, radiological and laboratory findings, and final diagnosis. Regarding the evaluation of nutritional status, the presence of wasting and kwashiorkor was assessed according to methods previously described by the WHO [7]. In particular, wasting was considered moderate or severe if weight-for-height z scores (WAZ) were below −2 or −3 standard deviations of the WHO Child Growth Standards median, respectively. Alternatively, the mid upper arm circumference (MUAC) was used and MUAC between 11.5 cm and 12.4 cm and <11.5 cm was deemed as an expression of moderate or severe wasting, respectively. Kwashiorkor was diagnosed in the presence of edema that was graded as mild, moderate, or severe if it affected only the feet, or the feet, legs, and upper limb, or it was generalized. In the final evaluation, cases with wasting or kwashiorkor were considered together and identified according to the degree of undernutrition as moderate acute undernutrition (MAU) or severe acute undernutrition (SAU, i.e., marasmus).

Children were categorized according to age (0–6 months, 7–12 months, 13–24 months, 25–59 months), gender, place of residence, main reason for hospitalization, and underlying comorbidities, including congenital heart disease, rheumatic heart disease, cerebral palsy, Down syndrome, and presence of HIV positivity. Place of residence was divided as follows: “near” included areas in close proximity to the hospital, “far” for other districts in Port Sudan, and “remote” was used for those who came from other states or from places that required more than one day’s travel.

### 2.2. Statistical Analysis

Descriptive statistics of the analyzed children were presented as frequencies and percentages to determine associations between predictors and outcome variables. For the main analysis, we used both bivariate and multivariate logistic regression analyses to report the association between undernutrition, cause of hospitalization, and their determinants. Crude and adjusted odds ratios (ORs) and 95% confidence intervals (ICs) were derived and two-sided *p*-values less than 0.05 were considered statistically significant.

## 3. Results

During the study period, a total of 1012 children (54.7% males) were hospitalized. Among them, 225 (22.2%) were less than 6 months old, 439 (43.3%) were between 6 months and 1 year, and 348 (34.4%) were between 13 and 59 months. For 92 (9.1%) patients, the provenance could not be ascertained; a total of 370 (36.6%) children came from districts considered near to the clinic, 456 (45.1%) from far areas, and 92 (9.1%) from remote regions.

In Figure 1, the main diagnoses at admission for the enrolled children are reported.

Lower respiratory tract infections (LRTIs) and acute gastroenteritis (AGE) were the most common diagnoses leading to hospitalization. LRTIs were diagnosed in 329 (32.5%) and AGE in 322 (31.8%) children. Parasitosis, mainly intestinal parasitosis due to *Entamoeba histolytica* and *Giardia lamblia*, was the third most common cause of hospital admission, accounting for 9.7% of all the cases. Of lower relevance as causes of hospitalization were upper respiratory tract infection (URTI; 7.1%), sepsis and septic shock (4.7%), urinary tract infection (UTI; 2.7%), malaria (2.4%), skin infections (1.8%), and other diseases including heart failure, epilepsy, and central nervous system infections. A total of 96 (9.5%) patients had an underlying condition, among which congenital heart disease (*n* = 39), HIV infection (*n* = 13), and Down syndrome (*n* = 7) were the most common. 

Acute undernutrition was evidenced in 493 (48.7%) children. Of them, only 16 (3.2%) were diagnosed with kwashiorkor. Among those with wasting, 74 (15.0%, of whom 56.8% were males) had moderate disease and 403 had severe disease (81.7%, of whom 64.7% were males). In the group aged 0–6 months, 86 infants (38.2%) were classified as SAU and 4 as MAU (1.8%), whereas most of these patients were well nourished. The rates of SAU and MAU were significantly higher among infants aged 7–12 months with a lower number of well-nourished patients (*p*-value < 0.05). A total of 262 children (59.7%) of this age group were classified as SAU, 40 (9.1%) as MAU, and 137 (31.2%) as well nourished. Similarly higher percentages of undernourished patients were found in the 13–24-month group, where 125 (52%) cases of SAU and 25 (10.4%) cases of MAU were detected (*p*-value < 0.01 vs. age group 0–6 months and <0.05 vs. age group 7–12 months). Lower, although higher than in the group of younger children, were the cases of SAU (*n* = 20; 18.5%) and MAU (*n* = 5; 4.6%) in children aged 2 to 5 years (*p*-value < 0.01 vs. age group 0–6 months and <0.05 vs. age group 7–12 months).

In Figure 2, the causes of hospitalization in children with SAU are reported. As shown, wasting or kwashiorkor were observed mainly in children with AGE (*n* = 249; 50.3%) or LRTI (*n* = 88; 17.8%). Parasitosis was diagnosed in 12.6% of children with SAU, sepsis in 4.7%, URTI in 3.4%, and skin infection in 1.2%.

Table 1 reports the potential associations between the demographic and clinical data of the enrolled subjects and the development of undernutrition.

Children with SAU had a higher prevalence of AGE (*p*-value < 0.05) and parasitosis (*p*-value < 0.05), whereas the prevalence of URTI and LRTI was higher in patients with no SAU. The risk of SAU and MAU was not influenced by gender as it was not different in females compared to males (OR 1.1; 95% CI, 0.84–1.44). On the contrary, age was strictly associated with undernutrition. Children aged 0–6 months were those with the lowest risk of undernutrition, whereas those aged 7–12 months were those with the greatest risk. In these patients, multivariate analysis revealed that SAU and MAU were 2.5 times (OR 2.51; 95% CI, 1.79–3.55) and 5.5 times (OR 5.56; 95% CI, 2.59–18.7) higher. The risk of both SAU and MAU remained significantly higher even in children aged 13–24 months, whereas it was significantly lower in patients 2 to 5 years old who appeared to be safe from severe malnutrition (OR 0.35; 95% CI, 0.19–0.63). The site of residence did not affect the risk of being malnourished, i.e., coming from other districts of Port Sudan compared to living next to the hospital or in other states. Patients with comorbidities did not have a statistically significant risk of SAU, not even those affected by congenital heart disease (CHD; OR 122; 95% CI, 0.71–2.10).

## 4. Discussion

The nutritional status of infants and preschool children is of paramount importance because it affects their lifetime health and intellectual capabilities. Nutrition is indeed a prominent topic on global agendas due to its far-reaching implications for human health, economic development, and social well-being [1]. Nutrition plays a fundamental role in human health at all stages of life. Adequate nutrition is essential for growth, immune function, cognitive development, and overall well-being. Conversely, malnutrition, including undernutrition and overnutrition, contributes to a range of health problems, including stunting, wasting, micronutrient deficiencies, obesity, and non-communicable diseases such as diabetes and cardiovascular diseases. Addressing malnutrition is crucial for improving health outcomes and reducing the burden of disease worldwide. Moreover, nutrition is closely linked to economic development and poverty reduction. Malnutrition, particularly among children, can have long-term effects on physical and cognitive development, leading to reduced productivity and earning potential in adulthood. This perpetuates cycles of poverty and inequality, hindering economic growth and social progress. Investing in nutrition interventions, such as promoting breastfeeding, improving access to nutritious foods, and providing micronutrient supplementation, can contribute to breaking the cycle of poverty and fostering sustainable development. In addition, nutrition is intricately linked to food security and sustainability. Ensuring access to an adequate and diverse diet is essential for meeting nutritional needs and promoting health. However, factors such as climate change, environmental degradation, food insecurity, and inequitable food distribution pose challenges to achieving nutrition security for all. Addressing these issues requires a holistic approach that considers the environmental, social, and economic dimensions of food production, distribution, and consumption. Nutrition is also a fundamental human right, as recognized by international agreements such as the Universal Declaration of Human Rights and the Convention on the Rights of the Child. Access to adequate nutrition is essential for realizing other human rights, including the right to health, education, and an adequate standard of living. However, disparities in access to nutritious foods, healthcare services, and socioeconomic opportunities perpetuate inequalities in nutrition outcomes, particularly among marginalized populations. Promoting equity in nutrition requires addressing underlying determinants of health and social justice. Nutrition intersects with several global health challenges, including infectious diseases, maternal and child health, and non-communicable diseases. Improving nutrition outcomes can contribute to achieving multiple health-related Sustainable Development Goals (SDGs), including SDG 2 (Zero Hunger), SDG 3 (Good Health and Well-being), and SDG 10 (Reduced Inequalities). Recognizing the interconnectedness of nutrition with other health and development priorities underscores its importance on global agendas. Given the multifaceted nature of nutrition and its profound impact on human health, economic development, and social equity, addressing malnutrition remains a top priority for global policymakers, practitioners, and advocates [1]. Collaborative efforts at the global, regional, and national levels are essential for implementing evidence-based interventions, strengthening health systems, and advancing policies that promote nutrition security and equitable access to nutritious foods for all.

The Global Nutrition Targets of the WHO for 2025 aim to reduce the number of stunted under-five children by 40%, and to maintain wasting in children to less than 5 years [8]. Data collected with this study show that in Sudan, particularly in the Red Sea State, this goal is far from being achieved. Compared to the data collected in 2018–2019 [5] that had shown a prevalence of moderate and severe wasting of 26.1% and 23.3%, respectively, this study found that, although the total number of children with acute undernutrition was quite similar (48.7% vs. 49.4%), the number of patients with SAU was significantly higher (81.7% vs. 23.3%). As the COVID-19 pandemic had its peak in the years immediately following the previous survey, we speculate that, at least in part, the increase in SAU cases among the children enrolled in this study may be ascribed to the pandemic itself. The large circulation of SARS-CoV-2 among children also seems to be suggested by the evidence that even children without SAU were frequently affected by a respiratory tract infection, the main clinical manifestation of COVID-19, with an incidence even greater than in children with SAU. On the other hand, vaccination coverage against COVID-19 in the general population of Sudan has been very low and many people including children were infected by SARS-CoV-2 [9]. Addressing the impact of COVID-19 on undernutrition among children under 5 years in Sudan requires coordinated efforts to strengthen food security, protect livelihoods, ensure access to essential healthcare services, and support nutrition interventions for vulnerable populations. Prioritizing nutrition-sensitive and nutrition-specific interventions, including social protection programs, food assistance, micronutrient supplementation, and breastfeeding support, can help mitigate the adverse effects of the pandemic on child nutrition outcomes in this country. Additionally, strengthening health systems and investing in resilient food systems are essential for building long-term resilience to future shocks and crises.

It is well known that undernutrition and infections are strictly related [10]. Undernutrition impairs immune system functions and favors infections and these, in turn, particularly when affects the gastrointestinal tract, worsen undernutrition [11]. Essential nutrients, including vitamins, minerals, and proteins, play key roles in supporting immune responses, such as the production of antibodies, cytokines, and immune cells. When individuals are undernourished, deficiencies in these nutrients weaken the immune system, compromising its ability to effectively recognize and combat pathogens. Gastrointestinal infections, including diarrheal diseases caused by pathogens such as bacteria, viruses, and parasites, further exacerbate undernutrition by exacerbating nutrient loss, reducing appetite, and impairing nutrient absorption. Moreover, gastrointestinal infections lead to fluid loss, electrolyte imbalances, and nutrient depletion, exacerbating existing nutritional deficiencies and impairing growth and development. The interplay between undernutrition and infections can have profound long-term consequences for health and development [10,11]. Breaking this cycle requires comprehensive interventions that address both undernutrition and infections, focusing on improving nutritional status, strengthening immune function, and preventing and treating gastrointestinal infections. Although we did not have data on biological markers related to the immune status of the study population, the linkage between undernutrition and infections is further highlighted by the evidence that most of the children admitted to the hospital suffered from AGE and the incidences of AGE and parasitosis were significantly lower in children without MAU or SAU. The prevention of infections, including COVID-19, through the administration of all the pediatric vaccines currently available remains essential to reduce the incidence of infections in undernourished children. Several studies have shown that, despite the dysfunction of both innate and adaptive immune system functions, malnourished children generally mount protective responses to all the pediatric vaccines [12,13]. Moreover, in our study population, a not marginal number of children suffered from intestinal parasitosis, and the prevalence of this condition was significantly higher in children with SAU than in those without. This seems to indicate that a high index of suspicion for amebiasis and giardiasis should be maintained in every reporting history of undernourished children with AGE, although the suggested systematic empirical treatment with metronidazole to face supposed amebiasis in acute undernourished children remains debatable [14]. On the contrary, instructions for mothers on preparing safe drinking and cooking water are essential to reduce the frequency of food contamination.

Various elements can affect the nutritional status of children including economic, social, and political factors [15]. It has been evidenced that the main role in conditioning the development of undernutrition is played by birth weight, maternal education, maternal age and nutrition, especially when considering pregnancy, maternal anemia, child’s birth order, child’s age, place of residence, toilet facilities, stool disposal systems, short periods of breastfeeding, and household income level [16,17,18,19]. Due to the limited information reported in the medical records used to perform this study, only gender, age, presence of comorbidities, and place of residence were evaluated as potential factors contributing to the development of MAU and SAU in children. Regarding gender, the results of previous studies are controversial as in some cases acute undernutrition was more common among males [20,21,22] and in other studies among females [23,24,25]. In this study, despite a slightly higher prevalence of acute undernutrition among boys, no relevant association with gender was found, with females and males having an equal risk of being affected by MAU and SAU. The role of gender remains undefined. Only further studies can clarify whether gender is important as a cause of undernutrition in children receiving the same power supply. Better defined seems to be the role of age. The results of this study clearly indicate that children in the first 6 months of life are at low risk of developing acute undernutrition. Breast feeding assures, in most cases, adequate nutritional intake. On the contrary, children aged 6–11 months and, although to a lower extent, those aged 7–24 months were those found at increased risk of acute undernutrition. The results of this study are congruent with those already published in several other developing countries [26,27,28] and quite like the UNICEF data collected in Sudan in 2014 showing the highest prevalence of SAU in 6–11-year-old children [2]. These findings suggest that when complementary feeding is introduced and the family cannot afford a high-quality diet, malnutrition can appear. Moreover, they indirectly highlight the importance of the mother’s education in conditioning the risk of acute undernutrition in infants and younger children. Interestingly, Abdel-Rahman et al. recently showed that only half of Sudanese mothers practiced an optimal early feeding practice, with important differences between regions in the country [29]. These findings suggest the need to develop breastfeeding promotion programs with consideration of regional variations and healthcare system interventions. Education affects a child’s nutritional status because educated mothers are more likely to know about adequate child nutrition and have more chances to be employed, increasing the purchasing power of the family for healthy food and access to health services [30].

Regarding the role of underlying morbidities as a cause of SAU and MAU, several studies have shown that children with chronic disease, including congenital heart disease, chronic kidney disease, or cystic fibrosis, even if they live in high-income countries, frequently suffer from nutrient deficiencies that are associated with the development of both acute and chronic undernutrition [31,32,33]. In this study, no relevant association between the presence of chronic underlying disease comorbidities and the risk of SAM was demonstrated. This finding is surprising as in children with chronic underlying disease various factors, such as increased caloric need, poor feeding tolerance, frequent vomiting, and a need for fluid limitations, reduce food intake and cause undernutrition [34]. However, it may be explained by the very low number of children with underlying disease enrolled in this study as well as by the fact that medical records are often incomplete, preventing any adequate statistical evaluation.

As far as the relevance of place of residence, it would be somewhat expected that children living nearby the hospital would be less affected by acute undernutrition than those living far from the hospital. The emergency hospital is accessible whenever, is completely free of charge, and health promotion regarding adequate child nutrition has been carried out in the past ten years. On the contrary, no difference was observed between children living next to the hospital or in other urban districts and those coming from rural places with no health services. This can be explained considering that the hospital was built in one of the poorest areas of Port Sudan, with precarious hygienic conditions, and that it is inhabited by a relevant number of members of the Beja tribe, who historically tend to be distrustful and rely on their own traditional diet and treatments.

This study has some limitations. First of all, it is a retrospective study. Moreover, the medical data of each child were not collected from electronic health record charts systematically reporting demographic, diagnostic, treatment, referral, and lifestyle information. Finally, the total number of enrolled children was relatively low. However, the reported information seems adequate to indicate that in the Red Sea State of Sudan, no reduction in acute malnutrition in children aged less than 5 years has recently occurred and that it is highly likely that the COVID-19 pandemic has played a relevant role in this regard, especially worsening the degree of undernutrition. In further studies, it would be interesting to compare the prevalence of undernutrition before the COVID-19 pandemic and during the pandemic to see if this is the real cause of the increase in the number of cases. Also, it would be interesting to compare with other areas in Sudan, as well as with the prevalence in other African countries or underdeveloped countries. Another correlation should be made with the food received by the patients included in the batch, which can be deficient both quantitatively and qualitatively.

## 5. Conclusions

This study shows that the area of Port Sudan is still suffering from an alarming prevalence of severe wasting and the risk of developing acute undernutrition seems strictly related to the introduction of complementary feeding and tends to reduce with increasing age. The measures already in place to prevent acute malnutrition should be reinforced with improvements in mother education on child feeding. Certainly, resources are essential and the consequences of refeeding syndrome should be considered in children presenting with marasmus and kwashiorkor. There must be continual nutritional and health assessments of these vulnerable individuals.

## Figures and Tables

**Figure 1 nutrients-16-00787-f001:**
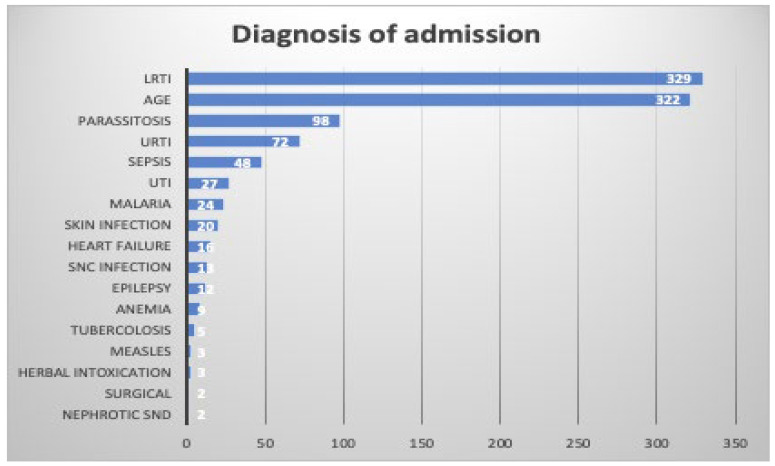
Diagnosis at admission among under-five children admitted to the Port Sudan Emergency Pediatric Hospital, Sudan. AGE, acute gastroenteritis; LRTI, lower respiratory tract infection; SNC, central nervous system; URTI, upper respiratory tract infection; UTI, urinary tract infection.

**Figure 2 nutrients-16-00787-f002:**
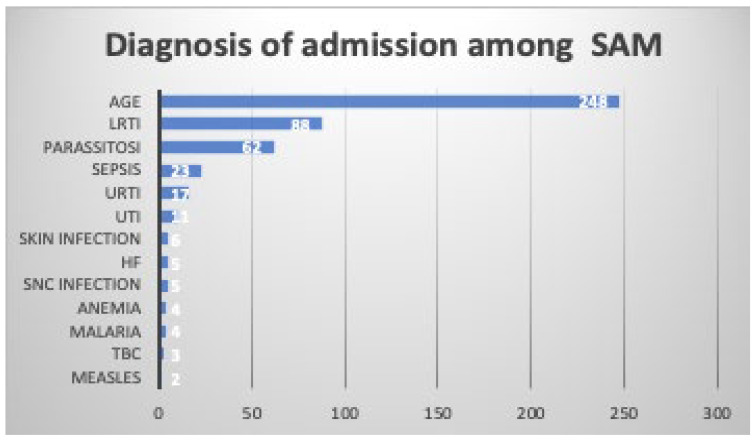
Diagnosis of admission among under-five children admitted to the Port Sudan Emergency Pediatric Hospital with severe undernutrition (SAU). AGE, acute gastroenteritis; HF, heart failure; LRTI, lower respiratory tract infection; SNC, central nervous system; URTI, upper respiratory tract infection; UTI, urinary tract infection.

**Table 1 nutrients-16-00787-t001:** Univariate and multivariate analysis of predictors of undernutrition among under-five children admitted to the Port Sudan Emergency Pediatric Hospital.

Predictors of SAU	Unadjusted OR	95% CI	*p* Value	Adjusted OR	95% CI	*p* Value
**Gender: female**	1.12	0.87–1.44	0.34	1.1	0.84–1.44	0.45
**Age**						
7–12 months	2.46	1.77–3.43	0	2.51	1.79–3.55	0
13–24 months	1.79	1.23–2.59	0	1.92	1.31–2.82	0
25–59 months	0.37	0.21–0.64	0	0.35	0.19–0.63	0
**Place of residence: far**	1.13	0.87–1.46	0.33	1.12	0.85–1.46	0.4
**Comorbidities**	1.43	0.92–2.25	0.11	1.39	0.87–2.25	0.16
CHD	1.24	0.75–2.07	0.39	1.22	0.71–2.10	0.47
SCD	0.08	0.04–0.12	0			
**Predictors of MAU**	Unadjusted OR	95% IC	*p* Value	Adjusted OR	95% IC	*p* Value
**Gender: female**	0.9	0.55–1.45	0.67	1.01	0.61–1.65	0.95
**Age**						
7–12 months	5.53	2.1–18.6	0	5.56	2.59–18.7	0
13–24 months	6.42	2.44–22	0	6.34	2.39–21.9	0
25–59 months	2.68	0.69–11	0.14	2.71	0.7–11.2	0.14
**Place of residence: far**	0.61	0.38–1	0.05	0.59	0.36–0.96	0.03
**Comorbidities**	0.28	0.04–0.92	0.08	0.29	0.04–0.98	0.09
CHD	0.18	0.01–0.86	0.09	0.19	0.01–0.91	0.1

CHD, congenital heart disease; CI, confidence interval; MAU, moderate acute undernutrition; OR, odds ratio; SAU, severe acute undernutrition; SCD, sickle cell disease.

## Data Availability

Data are contained within the article.

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
