# Peer review of "Determinants of Undernutrition among Children Admitted to a Pediatric Hospital in Port Sudan, Sudan"

_nutrients, 2024, doi:10.3390/nu16060787_

Round 1

Reviewer 1 Report

Comments and Suggestions for Authors I think it would have been interesting to compare the prevalence of undernutrition before the COVID19 pandemic and during the pandemic to see if this is the real cause of the increase in the number of cases. Also, it would have been interesting to compare with other areas in Sudan, as well as with the prevalence in other African countries or underdeveloped countries, especially since at References you also studied articles about malnutrition from other underdeveloped countries. Another correlation should be made with the food received by the patients included in the batch, which can be deficient both quantitatively and qualitatively. Comments on the Quality of English Language

Minor editing of English language required

Author Response

I think it would have been interesting to compare the prevalence of undernutrition before the COVID19 pandemic and during the pandemic to see if this is the real cause of the increase in the number of cases. Also, it would have been interesting to compare with other areas in Sudan, as well as with the prevalence in other African countries or underdeveloped countries, especially since at References you also studied articles about malnutrition from other underdeveloped countries. Another correlation should be made with the food received by the patients included in the batch, which can be deficient both quantitatively and qualitatively.

Re: Thank you very much for your suggestions. We included your comments in the revised manuscript.

Comments on the Quality of English Language

Minor editing of English language required

Re: The manuscript has been revised by a native English speaker.

Reviewer 2 Report

Comments and Suggestions for Authors

General Comments:

The absence of line numbers makes reviews and targeted suggestions difficult.  The journal should “fix” this issue.

The authors address a critical issue, namely undernutrition and malnutrition in a specific geographic area of Sudan.  The manuscript is reasonably well-written.  Interventions should be extended beyond maternal education…resources are essential…and refeeding syndrome consequences must be considered in this manuscript, especially those associated with children presenting marasmus and kwashiorkor.  There must be continual nutritional and health assessments of these vulnerable individuals.

Abstract:

This section is generally acceptable.  

A subsequent clinical presentation is missing…”main reason for hospitalization, and underlying _____ “ (some clinical malady belongs here…maybe comorbidities).

Introduction:

Page 1

1st paragraph: when discussing the first periods of life…while the statements are true, the key time is the first 1000 days of life.  This should be noted in the manuscript.

2nd paragraph: good introduction of Marasmus and Kwashiorkor.  Some overt presentations of these conditions should be noted.  See work by Cicely D. Williams and others.  Authors should consider cachexia and anorexia among those presenting undernutrition and malnutrition.  In addition, the authors should consider secondary malnutrition and associated maladies.

Page 2

3rd paragraph: The recent WHO report on malnutrition deserves greater coverage in this paragraph.

Methods:

1st paragraph: the approach outlined in this paragraph is reasonable. The dx of marasmus should be included in the diagnosis criteria.  Did the authors have approval from the hospital IRB?  This is critical to note.

2nd paragraph:  the text is reasonable

Page 3

Statistics:  reasonable stat approaches.

Results:

Figure 1 – good representation of dx; are there any hx data on the hospitalized children?  Any blood chemistries or anthropometric data?

Bottom paragraph:  good summary of pathologies

Page 4

1st paragraph:  authors should be certain to define SAU and MAU when initially presented in the ms

Figure 2 – need a graphic for MAU

Table 1 – good simple tabulation of demographics

Page 5

Bottom paragraph: good statement on SAU and MAU

Page 6

Discussion

1st paragraph: Good points of discussion; greater impact if there were a few photographs (carefully protecting the children’s identity) to emphasize the seriousness of the issues discussed in this ms

Were the children dx as COVID positive, or are the authors speculating?  Do the authors have any immunological biomarkers among these children? It seems the authors are relying on other publications for comments on COVID in pediatric populations.

2nd paragraph: Indeed immunological competence is compromised among undernourished and malnourish children (as well as adults); the comments would be strengthened if there were biological markers on the immune status of the children in this ms

3rd paragraph: good comments on factors that contribute to nutritional status; when mentioning breastfeeding, what is the frequency and duration of breastfeed in the Red Sea State of Sudan.  See interesting publication: Abdel-Rahman, M.E., El-Heneidy, A., Benova, L. et al. Early feeding practices and associated factors in Sudan: a cross-sectional analysis from multiple Indicator cluster survey. Int Breastfeed J 15, 41 (2020). https://doi.org/10.1186/s13006-020-00288-7

Page 7

If medical records are incomplete, what actions/recommendations should be implemented to better assess nutrition, health, family, and environmental issues?

If there is a lower risk of undernutrition during the first 6 mos of life, are these children breastfed?

Good comments on study limitations.  Unfortunately, a plethora of data were unavailable.

Page 8

Concluding comments should be expanded beyond material education on child feeding.

Author Response

General Comments:

The absence of line numbers makes reviews and targeted suggestions difficult.  The journal should “fix” this issue.

Re: We used journal’s format and line numbers are included.

The authors address a critical issue, namely undernutrition and malnutrition in a specific geographic area of Sudan.  The manuscript is reasonably well-written.  Interventions should be extended beyond maternal education…resources are essential…and refeeding syndrome consequences must be considered in this manuscript, especially those associated with children presenting marasmus and kwashiorkor.  There must be continual nutritional and health assessments of these vulnerable individuals.

Re: Thank you for your suggestions. Your comments have been added in the manuscript (p. 8).

Abstract:

This section is generally acceptable. 

A subsequent clinical presentation is missing…”main reason for hospitalization, and underlying _____ “ (some clinical malady belongs here…maybe comorbidities).

Re: Revised as suggested (p. 1).

Introduction:

Page 1

1st paragraph: when discussing the first periods of life…while the statements are true, the key time is the first 1000 days of life.  This should be noted in the manuscript.

Re: Clarified (p. 1).

2nd paragraph: good introduction of Marasmus and Kwashiorkor.  Some overt presentations of these conditions should be noted.  See work by Cicely D. Williams and others.  Authors should consider cachexia and anorexia among those presenting undernutrition and malnutrition.  In addition, the authors should consider secondary malnutrition and associated maladies.

Re: Added (p. 2)

Page 2

3rd paragraph: The recent WHO report on malnutrition deserves greater coverage in this paragraph.

Re: Added (p. 2).

Methods:

1st paragraph: the approach outlined in this paragraph is reasonable. The dx of marasmus should be included in the diagnosis criteria.  Did the authors have approval from the hospital IRB?  This is critical to note.

Re: Clarified (p. 3). Details on IRP approval are reported in the specific section of the manuscript (p. 8).

2nd paragraph:  the text is reasonable

Re: Thanks.

Page 3

Statistics:  reasonable stat approaches.

Re: Thank you.

Results:

Figure 1 – good representation of dx; are there any hx data on the hospitalized children?  Any blood chemistries or anthropometric data?

Re: No other data are available.

Bottom paragraph:  good summary of pathologies

Re: Thanks.

Page 4

1st paragraph:  authors should be certain to define SAU and MAU when initially presented in the ms

Re: Defined in the Methods (p. 3).

Figure 2 – need a graphic for MAU

Re: We think that the Table better summarizes MAU data and differences with SAU.

Table 1 – good simple tabulation of demographics

Re: Thank you.

Page 5

Bottom paragraph: good statement on SAU and MAU

Re: Thank you.

Page 6

Discussion

1st paragraph: Good points of discussion; greater impact if there were a few photographs (carefully protecting the children’s identity) to emphasize the seriousness of the issues discussed in this ms

Re: Unfortunately, we are not allowed to publish photographs.

Were the children dx as COVID positive, or are the authors speculating?  Do the authors have any immunological biomarkers among these children? It seems the authors are relying on other publications for comments on COVID in pediatric populations.

Re: Clarified (p. 6).

2nd paragraph: Indeed immunological competence is compromised among undernourished and malnourish children (as well as adults); the comments would be strengthened if there were biological markers on the immune status of the children in this ms

Re: Clarified (p. 6).

3rd paragraph: good comments on factors that contribute to nutritional status; when mentioning breastfeeding, what is the frequency and duration of breastfeed in the Red Sea State of Sudan.  See interesting publication: Abdel-Rahman, M.E., El-Heneidy, A., Benova, L. et al. Early feeding practices and associated factors in Sudan: a cross-sectional analysis from multiple Indicator cluster survey. Int Breastfeed J 15, 41 (2020). https://doi.org/10.1186/s13006-020-00288-7

Re: The suggested study has been added (p. 7).

 Page 7

If medical records are incomplete, what actions/recommendations should be implemented to better assess nutrition, health, family, and environmental issues?

Re: Your comment has been included in the text (p. 8).

If there is a lower risk of undernutrition during the first 6 mos of life, are these children breastfed?

Re: Written in the text (p. 7).

Good comments on study limitations.  Unfortunately, a plethora of data were unavailable.

Re: Thanks.

Page 8

Concluding comments should be expanded beyond material education on child feeding.

Re: Done (p. 9)

Reviewer 3 Report

Comments and Suggestions for Authors

The article, titled "Determinants of Undernutrition among Children Admitted in a Paediatric Hospital in Port Sudan, Sudan," examines the prevalence and factors contributing to acute undernutrition in children in Port Sudan after the COVID-19 pandemic.

Introduction: Provides a well-structured background on the issue of undernutrition, particularly in the context of Sudan. In my opinion the authors should reformulate the purpose of the study. It is not possible to assess the prevalence of malnutrition in all districts in Red Sea State on the basis of an analysis of the population residing in one medical facility. Methodologically, the main concern is the generalizability of the results due to the sample being limited to one hospital.

Methods: Describes the study design, data collection, and statistical analysis. The methods are clearly articulated, but there might be concerns about the representativeness of the sample (only one hospital's records were used) which could impact the study's generalizability. However, I have doubts (the purpose of the study) that the study is not done "after" the covid-19 pandemic the indicated period is February 1st, 2021, to January 31st, 2022 - ongoing pandemic ....

Results: Presents detailed findings on the prevalence of undernutrition and its associations with various factors such as age, gender, and place of residence.  The readability of the figures must be improved; their quality is low.

Discussion: This section effectively ties the results to the broader context of undernutrition in Sudan and globally. It discusses potential implications and limitations of the study. However, the discussion could benefit from a more detailed exploration of the study's findings in relation to existing literature.

Conclusion: Unfortunately, at present, the conclusions do not answer the purpose of the study.

English: Overall, the article is well-written with minor language mistakes  

Author Response

The article, titled "Determinants of Undernutrition among Children Admitted in a Paediatric Hospital in Port Sudan, Sudan," examines the prevalence and factors contributing to acute undernutrition in children in Port Sudan after the COVID-19 pandemic.

Re: Thank you for your suggestions. We revised the text according to your comments and those received from the other reviewers.

Introduction: Provides a well-structured background on the issue of undernutrition, particularly in the context of Sudan. In my opinion the authors should reformulate the purpose of the study. It is not possible to assess the prevalence of malnutrition in all districts in Red Sea State on the basis of an analysis of the population residing in one medical facility. Methodologically, the main concern is the generalizability of the results due to the sample being limited to one hospital.

Re: We clarified that we collected data in the area of Port Sudan (pp. 1 and 2). We included this among the study limitations (p. 8).

Methods: Describes the study design, data collection, and statistical analysis. The methods are clearly articulated, but there might be concerns about the representativeness of the sample (only one hospital's records were used) which could impact the study's generalizability. However, I have doubts (the purpose of the study) that the study is not done "after" the covid-19 pandemic the indicated period is February 1st, 2021, to January 31st, 2022 - ongoing pandemic ....

Re: The purpose of the study has been improved (pp. 1-2) and its limitations considered (p. 8). In addition, we clarified that the comment on COVID-19 pandemic is a speculation (p. 6).

Results: Presents detailed findings on the prevalence of undernutrition and its associations with various factors such as age, gender, and place of residence.  The readability of the figures must be improved; their quality is low.

Re: Reviewer 2 considered clear our Figures and the Table. For this reason, we did not change them.

Discussion: This section effectively ties the results to the broader context of undernutrition in Sudan and globally. It discusses potential implications and limitations of the study. However, the discussion could benefit from a more detailed exploration of the study's findings in relation to existing literature.

Re: A more detailed exploration of the study's findings in relation to existing literature and study limitations has been added (pp. 7-8).

Conclusion: Unfortunately, at present, the conclusions do not answer the purpose of the study.

Re: With the revision in the Introduction, we think that the Conclusions answer the purpose of the study.  

English: Overall, the article is well-written with minor language mistakes.

Re: The text has been revised by a native English speaker. 

Round 2

Reviewer 2 Report

Comments and Suggestions for Authors

The revised manuscript is good and provides important information about the nutrition issues in the Sudan.

Reviewer 3 Report

Comments and Suggestions for Authors

Thank you for resubmitting your work. At the moment I see that all suggested corrections have been made. I consider the paper now suitable for publication.